# Differential contribution of PBP occupancy and efflux on the effectiveness of β-lactams at their target site in clinical isolates of *Neisseria gonorrhoeae*

**Silvia López-Argüello**[1], **Eva Alcoceba**[1], **Paula Ordóñez**[1], **Biel Taltavull**[1,2], **Gabriel Cabot**[1,2], **Maria Antonia Gomis-Font**[1,2], **Antonio Oliver**[1,2], **Bartolome Moya**[1,2]*

**1** Servicio de Microbiología and Unidad de Investigación, Hospital Universitario Son Espases, Health Research Institute of the Balearic Islands (IdISBa), Palma, Spain, **2** Centro de Investigación Biomédica en Red en Enfermedades Infecciosas (CIBERINFEC), Palma, Balearic Islands, Spain

* bartolome.moya@ssib.es

**Data Availability Statement:** All data included in this manuscript is available through Mendeley Data repository as: Moya, Bartolome; Lopez, Silvia

## Abstract

*Neisseria gonorrhoeae* exhibits alarming antibiotic resistance trends and poses a significant challenge in therapeutic management. This study aimed to explore the association of *penA* alleles with penicillin-binding protein (PBP) occupancy patterns and reduced outer membrane permeability, impacting susceptibility to last-line cephalosporins and potential β-lactam candidates. The whole genome sequence, the MICs and PBP $IC_{50}$s were determined for 12 β-lactams and β-lactamase inhibitors in 8 clinical isolates with varying β-lactam sensitivity, 2 ATCC, and 3 WHO cephalosporin-resistant reference strains. The genetic analysis identified diverse determinants of β-lactam resistance including *penA*, *ponA*, *porB*, and *mtrR* alterations. Mosaic *penA* alleles were confirmed to be key determinants of cephalosporin resistance, with notable impacts on PBP2 $IC_{50}$ affinities (in the presence of all PBPs). Substitutions in positions V316 and A501 exhibited significant effects on β-lactam PBP2 occupancy and MICs. PBP1 inhibition showed marginal effect on β-lactam sensitivity and PBP3 acted as a sink target. Ertapenem and piperacillin emerged as potential therapies against cephalosporin-resistant *N. gonorrhoeae* strains, along with combination therapies involving tazobactam and/or efflux inhibitors. The study determined the β-lactam PBP-binding affinities of last-line cephalosporins and alternative β-lactam candidates in strains carrying different *penA* alleles for the first time. These findings provide insights for developing new antimicrobial agents and enhancers against emerging resistant strains. Further research is warranted to optimize therapeutic interventions for cephalosporin-resistant *N. gonorrhoeae* infections.

## Author summary

*Neisseria gonorrhoeae* is showing increasing antibiotic resistance and is a significant challenge to treat. This study looked at the connection between the genetic variability of the

(2024), "PBP occupancy and efflux of β-lactams in clinical isolates of Neisseria gonorrhoeae", Mendeley Data, V1, doi: 10.17632/38zzxh3kms.1 Whole genome sequencing data for the NG 3, NG 7, NG 12, NG 14, NG 19, NG 20, and NG 21 isolates are available through the European Nucleotide Archive (ENA) accession numbers ERS18422475, ERS18422476, ERS18422477, ERS18422478, ERS18422479, ERS18422480, ERS18422481 and ERS18422482.

**Funding:** Open Access was funded by "Programa LIBERI 2024" from IdISBa. BM received funds from RADIX17/3-1 fellowship and RADIX17/3-2 grant, program within the FUTURMed project IdISBa Research Institute of Health Sciences of the Balearic Islands, Hospital Universitario Son Espases, Palma, Spain. Sustainable Tourism Tax, Govern de les Illes Balears, by the Miguel Servet Research Contract Program CP20/00138 from the National Institute of Health Carlos III (ISCIII) and by the Agencia Estatal de Investigación (AEI - State Research Agency), Spain through the Plan Estatal de Investigación Científica PROYECTOS DE I+D+i PID2020-112654RB-I00/AEI/10.13039/501100011033. The assay development part of this work was supported by the award R01AI136803 to BM from the National Institute of Allergy and Infectious Diseases (NIAID). AO received funds from the by the Ministerio de Economía y Competitividad of Spain, Instituto de Salud Carlos III– co-financed by European Regional Development Fund "A way to achieve Europe" ERDF, through the Spanish Network for the Research in Infectious Diseases (RD16/0016). The content of this paper is solely the responsibility of the authors and does not necessarily represent the official views of the National Institute of Allergy and Infectious Diseases, the National Institutes of Health (NIH). The funders had no role in study design, data collection and analysis, decision to publish, or preparation of the manuscript.

**Competing interests:** AO has received fees as speaker and/or research grants from MSD, Pfizer and Wockhardt.

main β-lactam target (penicillin binding protein 2; PBP2), target accessibility, and susceptibility to last-line antimicrobial class as well as potential new candidates. Genetic analysis found various factors contributing to β-lactam resistance affecting drug targets, entry and efflux. Specific substitutions in PBP2 were confirmed as the key determinant of cephalosporin resistance with notable impacts on drug sensitivity. Ertapenem and piperacillin emerged as potential therapies against resistant strains along with combination therapies involving tazobactam or efflux inhibitors. The findings provide insights for developing new antimicrobial agents against emerging resistant strains while further research is needed for better therapeutic interventions for these infections.

## Introduction

With 82 million new cases reported annually, the obligatory human pathogen *Neisseria gonorrhoeae* is a serious and rapidly expanding global public health concern. In 2022, cases of gonorrhea rose by an even more concerning 33% compared to the 18% increase in previous year. The most recent ECDC bulletin states that beginning in 2022 and continuing through 2023, there were reported increases in gonorrhoea notifications, with a focus on young heterosexual individuals [1,2].

*N. gonorrhoeae*, naturally capable of genetic transformation, has been able to survive in the presence of antibiotics within various anatomical sites of its host. The utilization of extended-spectrum cephalosporins (ESCs) and azithromycin as recommended therapies has resulted in the emergence of extensive drug resistance (XDR) clones against every major antibiotic class. This resistance has historically evolved alongside every significant advancement in the field of antibiotic development [3,4].

Sporadic reports involving three strains of ceftriaxone-resistant *N. gonorrhoeae* were made between 2009 and 2013 which were a cause for public health concern. H041 was isolated in japan in 2009 [5]. Strain F89 was first discovered in France in 2010 and subsequently isolated in Spain [3,6]. Strain A8806, was isolated in Australia towards the end of 2013 [7]. However, no additional reports of H041, F89 or A8806 have been made. To track antibiotic resistance in *N. gonorrhoeae*, surveillance systems were implemented along with increased monitoring efforts in the upcoming years. Despite ongoing reports of decreased susceptibility, strains fully resistant to ceftriaxone remained relatively uncommon [8,9].

However, recent reports have indicated treatment failures when using ceftriaxone in combination with azithromycin, the last-line combination treatment, due to extensively drug-resistant (XDR) *N. gonorrhoeae* strains showing high-level resistance to azithromycin, as well as resistance to ceftriaxone, cefixime, and cefotaxime.

Actually, since 2019 some countries like the United Kingdom have advocated for a single high dose of ceftriaxone monotherapy to prevent an additional rise in azithromycin resistance [10]. This could potentially have severe implications for antibiotic stewardship in sexual health.

The development of resistance to ceftriaxone and other ESCs is a complex process, involving various step-wise mechanisms. [5]. Specifically, resistance determinants impacting first-line treatments consist of different mutated loci that can be transmitted through homologous recombination, in a specific sequence. (i) The initial stage in ESCs resistance is *penA*, which encodes penicillin-binding protein 2 (PBP2; the primary gonococcal target) [3]. High levels of ESCs resistance require the presence of the mosaic *penA* gene, which entails 60–70 amino acid alterations [5,11]. (ii) Mutations in *mtrR* resulting in changes to either the MtrR repressor or

the overlapping promoters of *mtrR* and *mtrCDE* [12], causing an upregulation in the production of the MtrCDE efflux pump affecting β-lactams and macrolides and tetracyclines among others [13]. (iii) Mutations in Porin PorB1b (*penB* determinant) that decrease the influx of antibiotics into the periplasmic space. This specific phenotype is observed only in strains that exhibit increased *mtrCDE* levels [14,15]. (iv) The final stage in ESCs resistance involves *ponA*, the second target for penicillin (PBP1). However, the addition of mutated *ponA* to a strain with the three preceding determinants has not resulted in any changes in the MIC. This could suggest the presence of epistasis or unidentified resistance determinants (such as *penC* or "Factor X") [5,16].

Single nucleotide polymorphisms affecting one or several alleles of the 23S rRNA, in conjunction with *mtrR*, are the primary mechanisms of resistance to macrolides [17].

Epistasis between *mtrR* transcriptional repressor and the *mtrCDE* pump (a hot spot of interspecific recombination) contributes to the resistance to azithromycin and selective maintenance. This represents a great complexity by which antibiotic resistance can arise through multiple-loci interactions up to the point that in some strains the full-length mosaic *mtrD* allele is required for high-level resistance [18]. However, in some strains, mutation of a single allele (23S rRNA) is sufficient to confer high-level azithromycin resistance and it can further develop under selection pressure [19].

Inactivation of efflux pumps in multidrug-resistant gonococcal isolates has demonstrated the ability to restore susceptibility to various antimicrobials, although this effect is not uniformly observed across genetically similar bacterial strains. Consequently, rationally optimized combinations including a specific efflux pump inhibitor (EPI) of the gonococcal *mtrCDE* efflux pump would be a future potential treatment option for gonorrhea. This could be a novel way to preserve antimicrobials armamentarium, and even reinstate no longer used antimicrobials, preventing the emergence of resistance [20].

*N. gonorrhoeae* possesses a relatively basic set of 5 PBPs, including PBP3 (*dacB*), PBP4 (*pbpG*), and *dacC* with carboxypeptidase activity [21,22]. The two high molecular mass transpeptidases, *ponA* (PBP1; Class A) and *penA* (PBP2; Class B) are both essential, with *penA* being the primary clinical target inhibited at significantly lower β-lactam concentrations than PBP1 in β-lactam-susceptible strains.

Reduced cephalosporin acylation by mosaic *N. gonorrhoeae* PBP2 is the main driver of cephalosporin high-level resistance. Mosaic variants display remarkable decreases in the acylation rate constants for its substrate analogs (β-lactam antibiotics) while maintaining necessary transpeptidase activity to sustain cell survival [3,5,6,23–26]. Targeting PBP2 is essential to exert substantial killing. However, some *ponA* alleles (PBP1) with a single substitution have been found to contribute to β-lactams resistance by significantly reducing its acylation efficiency [27]. On the other hand, PBP3, which is unrelated to antimicrobial resistance, shows the greatest level of expression in gonococcal membranes [22]. Nonetheless, this PBP may have a significant role as a sink target; where β-lactam molecules binding to PBP3 become inaccessible for reaching their essential targets (i.e. PBP1 and 2) [28]. No mosaic variants have been discovered in PBP1 and PBP3, unlike the three mosaic PBP genes found in β-lactam resistant streptococci [29]. The emergence of mosaicism may be dependent on the existence of genes encoding PBPs with low affinity for β-lactams within the gene pool of *Neisseria* commensal species [18].

The efficacy of β-lactams in *N. gonorrhoeae* is not only affected by their selectivity towards PBPs and their acylation rate constants but also permeation through porins and susceptibility to efflux [30]. The second-order rate constant of acylation ($k_2/K_s$) of β-lactams has been extensively studied via fluorescence anisotropy assays for the primary target *ponA* following cloning and purification of a C-terminal domain construct of PBP2 from resistant strains. Although

we have recently published an extensive dataset on PBP-binding $IC_{50}$s for two type strains [31], there is currently no study available on membrane preparations of resistant isolates that considers the relative abundance of each PBP in *N. gonorrhoeae*. Fluorescence anisotropy assays and crystallization can be utilized to calculate the micro-constants of binding, while the isolated membrane fractions PBP-binding assay concurrently determines the binding for all PBPs [30,32–35]. This approach closely mimics the physiological conditions at the periplasmic target site of bacteria. The combination of both assays with permeability data provides a realistic and comprehensive understanding of PBP binding and target site penetration that can help filling the substantial gaps on the mechanistic basis to enhance β-lactam-based therapy for this important pathogen [36].

Therefore, in this research, we aimed to assess the PBP occupancies of clinical strains exhibiting reduced sensitivity to first-line cephalosporins as well as other chemically diverse β-lactams that may be clinically relevant. Our goal was to establish a correlation between these measurements and their efficacy and tolerance towards efflux pumps. To achieve this, we conducted genetic and phenotypic characterizations on eight *N. gonorrhoeae* clinical isolates (selected based on their sensitivities to cephalosporins and/or azithromycin) isolated from Hospital Universitario Son Espases and Hospital Clínic de Barcelona between 2016 and 2023. We determined their minimum inhibitory concentrations in the presence or absence of resistant *penA* alleles, *mtrD* inactivation and efflux pump inhibitors, as well as their PBP $IC_{50}$s in membrane preparations. In addition, for a more comprehensive analysis, we also characterized two wild type ATCC 19424 and ATCC 49226 strains, along with WHO X (H41), WHO Y (F89), and WHO Z (A8806) cephalosporin-resistant reference strains.

## Results

### Genetic characterization and antimicrobial susceptibility

Eight clinical isolates, two ATCC and three WHO resistant reference strains with increasing β-lactam MICs were selected for this study (Table 1). The strains were tested for penicillin (0.008–4 mg/L), ceftriaxone (0.002–2 mg/L), cefixime (0.004–4 mg/L), and cefotaxime (0.004–8 mg/L) susceptibility. None of the studied strains possessed any of the plasmidic penicillin determinant $bla_{\text{TEM-1 or}}$ $bla_{\text{TEM-135}}$ variants. The two reference strains, ATCC 19424 and ATCC 49226, showed the XV and the XXII types, respectively, for *penA* (PBP2), the major β-lactam resistance determinant (Fig 1). Both strains were sensitive to all tested cephalosporins (MICs ≤ 0.002–0.008 mg/L and < 0.002–0.064 mg/L, respectively; Table 1). ATCC 19424 carried wild-type alleles for all β-lactam resistance-related genes and was the only strain sensitive to penicillin. ATCC 49226 carried the A39T mutation on *mtrR*, resulting in a penicillin MIC of 0.5 mg/L (S1 Table).

*N. gonorrhoeae* isolates with reduced susceptibility to β-lactams including extended spectrum cephalosporins are associated with alterations in PBP2, PBP1, PorB, MtrR, and *mtrR* promoter region. Isolates NG 7 and NG 20 had type II *penA* alleles, expressed PorB1b porins with the G120K, A121N and A121S substitutions, and had a *N. meningitidis*-like (NM-like) promoter and a G45D change in the efflux pump regulator *mtrR* respectively (S1 Table). Both isolates showed slight MIC increases for penicillin (0.25–0.5 mg/L), cefixime and cefotaxime (0.016–0.031 mg/L) (Table 1).

NG 19 was found to have a PBP2 type V variant and a L421P change in PBP1 (*ponA*). It expressed PorB1b porin with G120K and A121N substitutions, and displayed a NM-like promoter, as well as an A39T substitution in *mtrR* (S1 Table). The only differential trait of this strain was a slight increase in ceftriaxone MIC, up to 0.008 mg/L.

**Table 1. Minimum inhibitory concentrations of β-lactam antibiotics and β-lactamase inhibitors in the studied *N. gonorrhoeae* strains.**

| Strain[a] | MIC of the indicated drug (mg/L)[b] | | | | | | | | | | | | | | | |
|---|---|---|---|---|---|---|---|---|---|---|---|---|---|---|---|---|
| | PenG | ETP | CFM | CTX | CRO | CAZ | AVI | CAZ/AVI[c] | TOL | PIP | TZ | TOL/TZ[c] | PIP/TZ[c] | CIP | TET | AZT |
| ATCC 19424 | 0.008 | 0.004 | 0.004 | 0.004 | 0.002 | 0.016 | 128 | 0.008 | 0.032 | <0.002 | 0.125 | <0.002 | <0.002 | 0.002 | 0.25 | 0.016 |
| ATCC 49226 | 0.5 | 0.008 | 0.016 | 0.008 | 0.004 | 0.032 | >256 | 0.064 | 0.25 | 0.125 | 1 | <0.002 | 0.064 | 0.004 | 1 | 0.25 |
| NG 3 | 0.5 | 0.004 | 0.032 | 0.125 | 0.064 | 0.5 | >256 | 0.5 | 0.125 | 0.064 | 2 | <0.002 | 0.032 | >32 | 4 | 0.25 |
| NG 7 | 0.25 | 0.002 | 0.032 | 0.016 | 0.004 | 0.032 | >256 | 0.064 | 0.5 | 0.25 | 2 | <0.002 | 0.032 | 0.016 | 2 | 16 |
| NG 12 | 1 | 0.032 | 0.25 | 0.25 | 0.125 | 1 | >256 | 0.5 | 8 | 0.125 | 16 | 2 | 0.25 | 8 | 4 | 0.5 |
| NG 14 | 1 | 0.032 | 0.125 | 0.25 | 0.125 | 1 | >256 | 0.5 | 2 | 0.032 | 16 | 2 | 0.125 | 16 | 4 | 0.5 |
| NG 19 | 0.5 | 0.004 | 0.008 | 0.032 | 0.008 | 0.125 | >256 | 0.125 | 0.125 | 0.032 | 1 | <0.002 | <0.002 | >32 | 4 | 1 |
| NG 20 | 0.5 | 0.008 | 0.016 | 0.032 | 0.004 | 0.125 | >256 | 0.125 | 0.5 | 0.125 | 2 | <0.002 | <0.002 | 0.008 | 1.5 | 2048 |
| NG 21 | 1 | 0.016 | 0.064 | 0.125 | 0.064 | 1 | >256 | 0.5 | 1 | 0.125 | 4 | <0.002 | 0.125 | >32 | 4 | 0.25 |
| NG 22 | 1 | 0.008 | 0.125 | 0.125 | 0.064 | 1 | >256 | 0.5 | 4 | 0.25 | 8 | <0.002 | 0.5 | >32 | 2 | 0.25 |
| WHO X | 4 | 0.032 | 4 | 8 | 2 | 16 | >256 | 16 | 64 | 0.125 | 32 | 64 | 0.25 | >32 | 2 | 0.25 |
| WHO Y | 4 | 0.002 | 4 | 4 | 1 | 16 | 32 | 16 | 128 | 0.032 | 4 | <0.002 | 0.032 | >32 | 4 | 0.25 |
| WHO Z | 2 | 0.016 | 2 | 4 | 0.5 | 4 | >256 | 4 | 16 | 0.125 | 64 | 32 | 0.25 | >32 | 4 | 0.25 |

[a] *N. gonorrhoeae* strains ATCC 19424 and ATCC 49226; clinical strains NG 3, NG 7, NG 12, NG 14, NG 19, NG 20, NG 21 from Hospital Universitario Son Espases (Spain) and NG 22 from Hospital Clínic de Barcelona (Spain); and WHO reference strains NCTC 13820 (WHO X), NCTC 13821 (WHO Y) and NCTC 13822 (WHO Z)

[b] Broth microdilution minimum inhibitory concentrations (MICs) were performed in accordance with the Clinical and Laboratory Standards Institute (CLSI) guidelines. The antibiotics tested were: penicillin G (PenG), ertapenem (ETP), cefixime (CFM), cefotaxime (CTX), ceftriaxone (CRO), ceftazidime (CAZ), ceftolozane (TOL), piperacillin (PIP), avibactam (AVI), tazobactam (TZ), ceftazidime/avibactam (CAZ/AVI), ceftolozane/tazobactam (TOL/TZ), piperacillin/tazobactam (PIP/TZ), and azithromycin (AZT)

[c] MICs for CAZ/AVI, TOL/TZ, and PIP/TZ were performed with a fixed concentration of the beta-lactamase inhibitor (BLI) avibactam or tazobactam (4 mg/L).

Isolates NG 3, 21, and 22 (MLST$_{ST7827}$, NG-STAR$_{ST38}$) exhibited type XIII *penA* alleles, the L421P PonA substitution, and expressed PorB1b porin with G120K and A121D substitutions, as well as a single A deletion in position -35 and a G45D change in the *mtrR* regulator (S1 Table). Phenotypically, the three isolates displayed significant higher MICs for penicillin (0.5–1 mg/L; p < 0.05) and ESCs (0.032–0.125 mg/L p < 0.005), just below the resistance break-point (EUCAST).

Isolates NG 12 and NG 14 (MLST$_{ST1901}$, NG-STAR$_{ST5569}$) carried the mosaic *penA* XXXIV allele variant, which correlated with significantly reduced sensitivity to ESCs at concentrations of 0.125–0.25 mg/L (p < 0.005) (**Table 1**).

As previously reported, the reference strains WHO X and WHO Z (MLST$_{ST7363}$) contained mosaic PBP2 alleles (WHO X = XXXVII, WHO Z = LXIV) [37]. These alleles confer resistance to ESCs (0.5–8 mg/L; p < 0.005) and penicillin. However, WHO Y (MLST$_{ST1901}$, NG-STAR$_{ST16}$) has a mosaic *penA* XLII variant, which resulted in ESC resistance, but penicillin (MIC = 1 mg/L) remained within the sensitivity range (**Table 1**).

The clinical isolates and resistant reference strains displayed low ertapenem MICs, ranging from 0.002 to 0.032 mg/L, even for the mosaic PBP2 carrying strains, which only experienced an 8-fold MIC change. Strain WHO Y exhibited an unusually low MIC 0.002 mg/L. In contrast, third generation cephalosporins (e.g. ceftriaxone) showed a 250- to 2000-fold increase in MIC values. However, the β-lactam/BLI combinations displayed a scattered range of MIC values: ceftazidime/avibactam ranged from 0.125 to 16 mg/L, ceftolozane/tazobactam ranged from 0.002 to 64 mg/L, and piperacillin/tazobactam had a narrower range of 0.002 to 0.5 mg/L. (**Fig 1**).

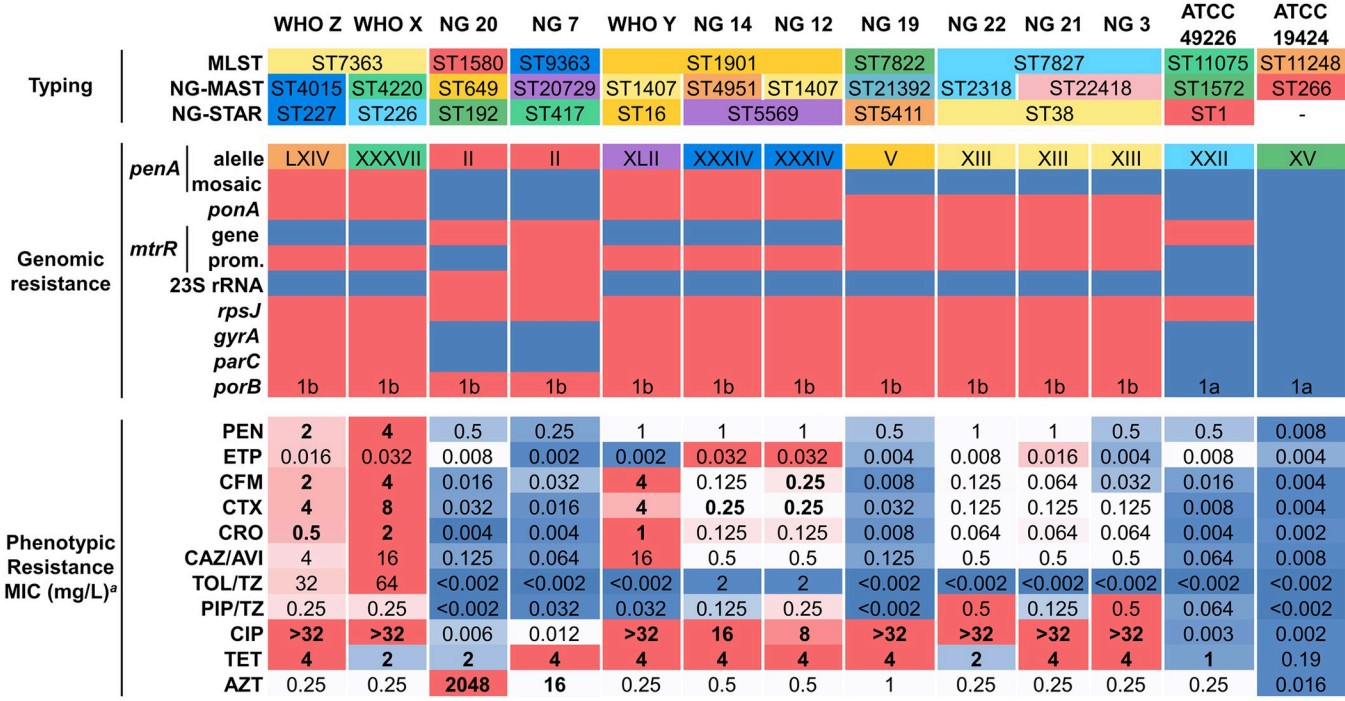

**Fig 1. Typing, genomic, and phenotypic antimicrobial resistance features of the 2 ATCC reference strains, 8 clinical strains, and the 3 WHO *N. gonorrhoeae* reference WHO X, WHO Y, and WHO Z strains.** Red indicates a resistant allele, while blue indicates a wild type allele. The MIC values are color-coded from dark blue (lower values) to red (higher values) for each drug. MIC values in bold indicate resistance based on EUCAST breakpoints. Clinical resistance breakpoints are not available for azithromycin, ertapenem, ceftazidime/avibactam, ceftolozane/tazobactam, and piperacillin/tazobactam. The sensitive breakpoint values according to EUCAST are: PEN ≤ 0.06, CFM ≤ 0.125, CTX ≤ 0.125, CRO ≤ 0.125, CIP ≤ 0.03, and TET ≤ 0.5. Azithromycin lacks clinical EUCAST breakpoints; therefore, EUCAST's epidemiological cut-off (ECOFF) value of MIC > 1 mg/L was used to define resistance. [a] Please refer to Table 1 for antibiotic abbreviations.

In relation to azithromycin resistance, only two strains (NG 7 and NG 20) exhibited clinical resistance as per the CLSI guidelines due to specific genetic mutations: C2611T in NG 7 (MIC = 16 mg/L) and A2059G in NG 20 (MIC = 2048 mg/L), along with additional mutations in *mtrR*, NM-like promoter for NG 7, and G45D for NG 20. (**S1 Table**). NG 7 and NG 20 were the only isolates, in addition to the two ATCC reference strains, that exhibited sensitivity to ciprofloxacin. In contrast, the remaining isolates and the WHO reference strains were resistant. (MIC range 8 to > 32 mg/L). The resistant isolates harbored specific quinolone resistance patterns (*gyrA* S91F and D95A/G/N; *parC* D86N, S87R and / or S88P).

In the total absence of plasmid mediated TetM in our strains, the combination of *rpsJ* V57M and *mtrR*-associated mutations (**S1 Table**) conferred resistance to tetracycline, with MICs ranging from 1 to 4 mg/L, in all studied strains except for ATCC 19424 (**Table 1**).

## Correlation of the PBP alleles with PBP IC$_{50}$ and antimicrobial susceptibility

This study determined the PBP IC$_{50}$ for cefixime, cefotaxime, ceftriaxone, ceftazidime, ceftolozane, piperacillin, avibactam, tazobactam, and the combinations ceftazidime/avibactam, ceftolozane/tazobactam, and piperacillin/tazobactam of each clinical isolate and reference strain using the Bocillin FL competition assay (**Fig 2**).

The selectivity of the drugs and combination was defined as the PBP that was inhibited with a drug concentration at least 4-fold lower than that of the next most inhibited PBP. As

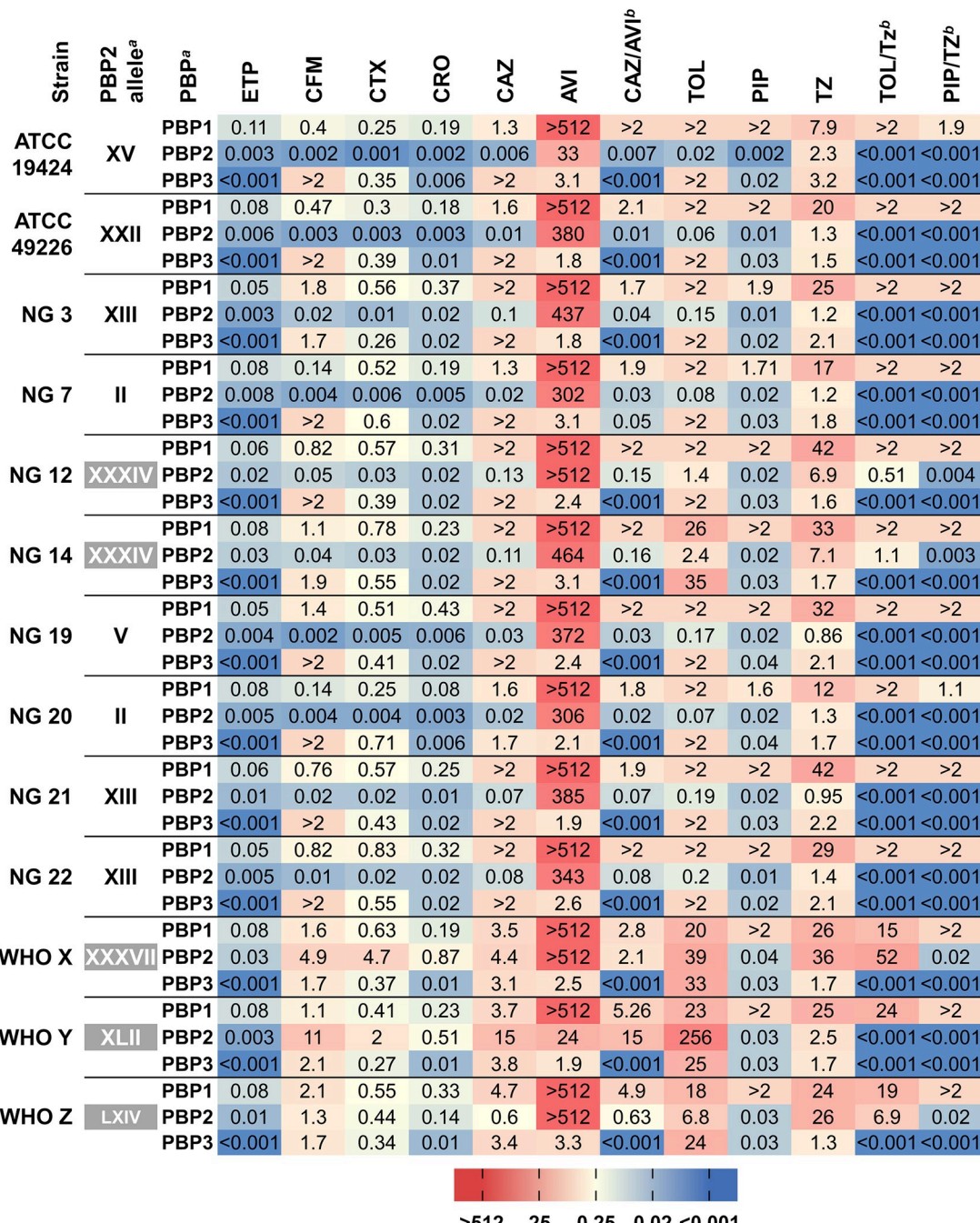

**Fig 2. Heatmap representing the PBP IC$_{50}$s of β-lactam antibiotics and β-lactamase inhibitors in the studied *N. gonorrhoeae* strains.** [a] PBP, penicillin-binding proteins. Mosaic alleles highlighted in grey. [b] a fixed concentration of the BLI avibactam or tazobactam was used (4 mg/L). The median values from 3 experiments are shown. Please see Table 1 for drug abbreviations. When the main PBP target was not inhibited using the regular concentrations (0.001 to 0.125 mg/L) an extended range (0.016 to 2 mg/L or 1 to 512 mg/L) was used.

previously determined [31], third generation cephalosporins (3GCs) were selective for PBP2, while the fifth generation ceftolozane half-maximally inhibited PBP3 or PBP2 and PBP3 depending on the strain. The BLI tazobactam was selective for PBP2, while avibactam maximally inhibited PBP3 (**Fig 2**). The combinations of beta-lactam with either BLI were selective

for PBP3 or PBP2 and PBP3. Due to their selectivity for the non-lethal target PBP3, ceftolozane, avibactam and its combination with ceftazidime showed the highest MIC values (**Table 1**).

First of all, it should be noted that the $IC_{50}$ values of mosaic PBP2 variants were significantly higher compared to those of non-mosaic variant-containing strains for 3GCs, (p < 0.005), fifth generation ESCs (p < 0.05) and β-lactam-BLI combinations (p < 0.005). Isolates NG 7, NG 19 and NG 20 and strain ATCC 49226, carrying PBP2 non-mosaic types II, V, and XXII showed a slight increase (3- to 7-fold) in cephalosporins PBP $IC_{50}$ (**Fig 2** and **S2 Table**) compared to the wild type ATCC 19424 (type XV). This led to low (2- to 8-fold) or non-apparent MIC increases for cefixime and ceftriaxone respectively (**Fig 1**).

Isolates NG 3, NG 21, and NG 22 (non-mosaic XIII PBP2) exhibited lower overall susceptibilities (8- to 32-fold) and demonstrated a 8- to 24-fold increase in the PBP2 $IC_{50}$s (ceftriaxone 0.013–0.018 mg/L, cefixime 0.013–0.018 mg/L, and cefotaxime 0.014–0.021 mg/L) of third-generation cephalosporins compared to the wild-type strain ATCC 19424 (0.001–0.002 mg/L). They also showed an approximately 12-fold increase in the PBP2 $IC_{50}$ of ceftazidime (0.073–0.1 vs 0.006 mg/L) and ceftolozane (0.15–0.2 vs 0.015 mg/L).

Isolates carrying the PBP2 type XXXIV mosaic variant (NG 12 and NG 14) elicited significantly (p < 0.005) higher PBP2 $IC_{50}$s for antigonococcal 3GCs (ceftriaxone 0.02 mg/L, cefixime 0.05 mg/L, and cefotaxime 0.03 mg/L) compared to wild-type values (0.001–0.002 mg/L), 16- to 20-fold increase in ceftazidime and 90- to 150-fold in ceftolozane $IC_{50}$s (1.4–2.4 mg/L) (**Fig 2**). This led to MIC increases of 16- to 64-fold for 3GCs and 64 to 256 times for ceftolozane compared to the wild-type strain ATCC 19424.

The isolates WHO X, WHO Y, and WHO Z, which are resistant to cephalosporins and carry mosaic PBP2 variants (type XXXVII, XLII, and LXIV), showed the highest $IC_{50}$ values (p < 0.005) for 3GCs (ceftriaxone 0.1–0.9 mg/L, cefixime 1.3–11 mg/L, cefotaxime 0.4–4.7 mg/L, and ceftazidime 0.6–15 mg/L). This represented a 75- to 5,000-fold decrease in PBP2 affinities, which was even lower for ceftolozane, 450- to 8,200-fold.

The $IC_{50}$s of ertapenem PBP2 exhibited minimal changes in accordance with MIC values. Isolates carrying PBP2 type II, V, XV and XXII variants (ATCC 19424, ATCC 49226, NG20, NG7, and NG19) displayed no significant changes (0.003–0.008 mg/L), while a marginal increase was observed in strains NG 3, NG 21 and NG 22 carrying type XIII variant (0.003–0.011 mg/L) (**Fig 2**). The mosaic PBP2 variants (XXXIV, XXXVII, XLII, and LXIV) exhibited only a 5- to 9-fold increase in $IC_{50}$ (0.013–0.026 mg/L) compared to the wild type strain 19424 (0.003 mg/L), with the exception of strain WHO Y, which showed an $IC_{50}$ of 0.003 mg/L.

The results of the combinations of β-lactam and BLIs varied. Avibactam 4 mg/L (PBP2 $IC_{50}$ = 23 to > 512 mg/L) in combination with ceftazidime reduced the PBP3 $IC_{50}$ values below the limit of detection (< 0.001 mg/L) in all tested strains. However, this combination did not affect the PBP2 $IC_{50}$ values or the MICs. Conversely, when 4 mg/L of tazobactam (PBP2 $IC_{50}$ = 0.9–2.5 mg/L) were added to ceftolozane, it resulted in a 15- to 128,000-fold PBP2 $IC_{50}$ reduction in all strains except for the NG 12, NG 14, WHO X and WHO Z strains carrying mosaic PBP2 alleles (XXXIV, XXXVII and LXIV) which showed a modest 2-fold maximum reduction. The MIC values were correspondingly decreased. However, when the BLI was added to piperacillin, with significantly lower MICs compared to ceftolozane, a PBP2 $IC_{50}$ reduction of 5- to 26-fold was observed in all strains except WHO X and WHO Z (XXXVII and LXIV). The addition of tazobactam to ceftolozane or piperacillin resulted in a significant $IC_{50}$s shift in strain WHO Y PBP2 (allele XLII) from 128 to <0.001 mg/L and from 0.026 to <0.001 mg/L, respectively. However, only ceftolozane showed a subsequent MIC reduction.

The PBP2 $IC_{50}$ values obtained in the presence of all PBP targets showed a strong direct correlation with MICs ($\rho$ = 0.85–0.99), except for piperacillin ($\rho$ = 0.22) and the combinations of

ceftazidime/avibactam ($\rho = 0.69$) and piperacillin/tazobactam ($\rho = 0.23$) (**Fig 3**). In addition, the piperacillin/tazobactam MIC/PBP2_IC$_{50}$ ratios ranged from 2 to 500 (**S3 Table**).

To ascertain whether the increasing resistance to ESCs was due to the different *penA* alleles we transformed the full-length *penA* allele from isolates with significantly reduced sensitivity to ESCs (and the cephalosporin resistant WHO X, Y and Z strains into the recipient strain FA1090. The experiments confirmed that the different *penA* alleles conferred increased resistance to all tested β-lactams (**Table 2**). Transformation of *penA* alleles into the recipient strains increased the recipient strain MICs of cefixime between 2- and 250-fold (MICs 0.016 to 2 mg/L) cefotaxime 8- to 500-fold (MICs 0.008 to 1 mg/L) and of ceftriaxone from 2- to 500-fold (MICs 0.004 to 1 mg/L) (**S4 Table**), which were above the resistance breakpoints for the transformants carrying WHO X, Y and Z *penA* alleles (9). Remarkably, all the transformants showed minimal ertapenem MIC changes, between 2- and 8-fold (MICs 0.004 to 0.016). Regarding BL-BLI combinations, ceftazidime/avibactam showed the most notable MIC increase between 4- and 250-fold (MICs 0.064 to 8 mg/L), changes in ceftolozane/tazobactam were only observed in NG 12, WHO X and Y strains (2-, 1000- and 500-fold respectively), however no MIC changes were documented for piperacillin/tazobactam.

In order to determine if the observed increase in resistance to extended-spectrum cephalosporins (ESCs) was attributable to the various *penA* alleles, we introduced the complete *penA* allele from isolates exhibiting markedly reduced sensitivity to ESCs, as well as from the cephalosporin-resistant WHO X, Y, and Z strains, into the recipient strain FA 1090. In every transformant analyzed, with the exception of *penA*$_{WHOY}$, the *penA* sequence that was introduced was found to be identical to that of the corresponding donor. In the case of the transformant *penA*$_{WHOY}$, two single nucleotide polymorphisms (SNPs) (A294G, G303T) were identified. However, these SNPs were synonymous, suggesting that this particular segment of the mosaic *penA* allele is unlikely to influence the minimum inhibitory concentrations of ESCs.

The introduction of *penA* alleles into the recipient strains resulted in a significant elevation of the MICs for cefixime, ranging from 2- to 250-fold (MICs 0.016 to 2 mg/L), for cefotaxime from 8- to 500-fold (MICs 0.008 to 1 mg/L), and for ceftriaxone from 2- to 500-fold (MICs 0.004 to 1 mg/L) (**Tables 2 and S4**). These values exceeded the resistance breakpoints for the transformants harboring the WHO X, Y, and Z *penA* alleles. Notably, all transformants

**Table 2. Changes in the susceptibilities of in vitro transformants of the receptor strain FA 1090 with different *penA* alleles.**

| Transformant FA 1090[a] | MIC of the indicated drug (mg/L)[b] | | | | | | | |
|---|---|---|---|---|---|---|---|---|
| | **PenG** | **ETP** | **CFM** | **CTX** | **CRO** | **CAZ/AVI** | **TOL/TZ** | **PIP/TZ** |
| receptor | 0.064 | 0.002 | 0.008 | 0.002 | 0.002 | 0.032 | <0.016 | <0.016 |
| *penA*$_{NG3}$ | 0.125 | 0.004 | 0.016 | 0.016 | 0.004 | 0.125 | <0.016 | <0.016 |
| *penA*$_{NG12}$ | 0.125 | 0.012 | 0.064 | 0.032 | 0.004 | 0.25 | 0.032 | <0.016 |
| *penA*$_{NG14}$ | 0.125 | 0.012 | 0.064 | 0.032 | 0.008 | 0.25 | 0.016 | <0.016 |
| *penA*$_{NG21}$ | 0.125 | 0.002 | 0.016 | 0.016 | 0.004 | 0.125 | <0.016 | <0.016 |
| *penA*$_{NG22}$ | 0.125 | 0.002 | 0.016 | 0.008 | 0.004 | 0.064 | <0.016 | <0.016 |
| *penA*$_{WHOX}$ | 0.5 | 0.016 | 2 | 1 | 1 | 4 | 16 | <0.016 |
| *penA*$_{WHOY}$ | 0.25 | 0.002 | 1 | 1 | 0.5 | 8 | <0.016 | <0.016 |
| *penA*$_{WHOZ}$ | 0.5 | 0.008 | 1 | 1 | 0.25 | 1 | 8 | <0.016 |

[a] Strain FA 1090 was transformed with the full-length PCR of the different *penA* alleles from clinical strains NG 3, NG 12, NG 14, NG 21 and NG 22 and WHO reference strains WHO X, WHO Y and WHO Z

[b] Agar microdilution MICs were performed following CLSI guidelines [38]. The MICs for CAZ/AVI, TOL/TZ, and PIP/TZ were determined using the ETEST method. Please see Table 1 for drug abbreviations.

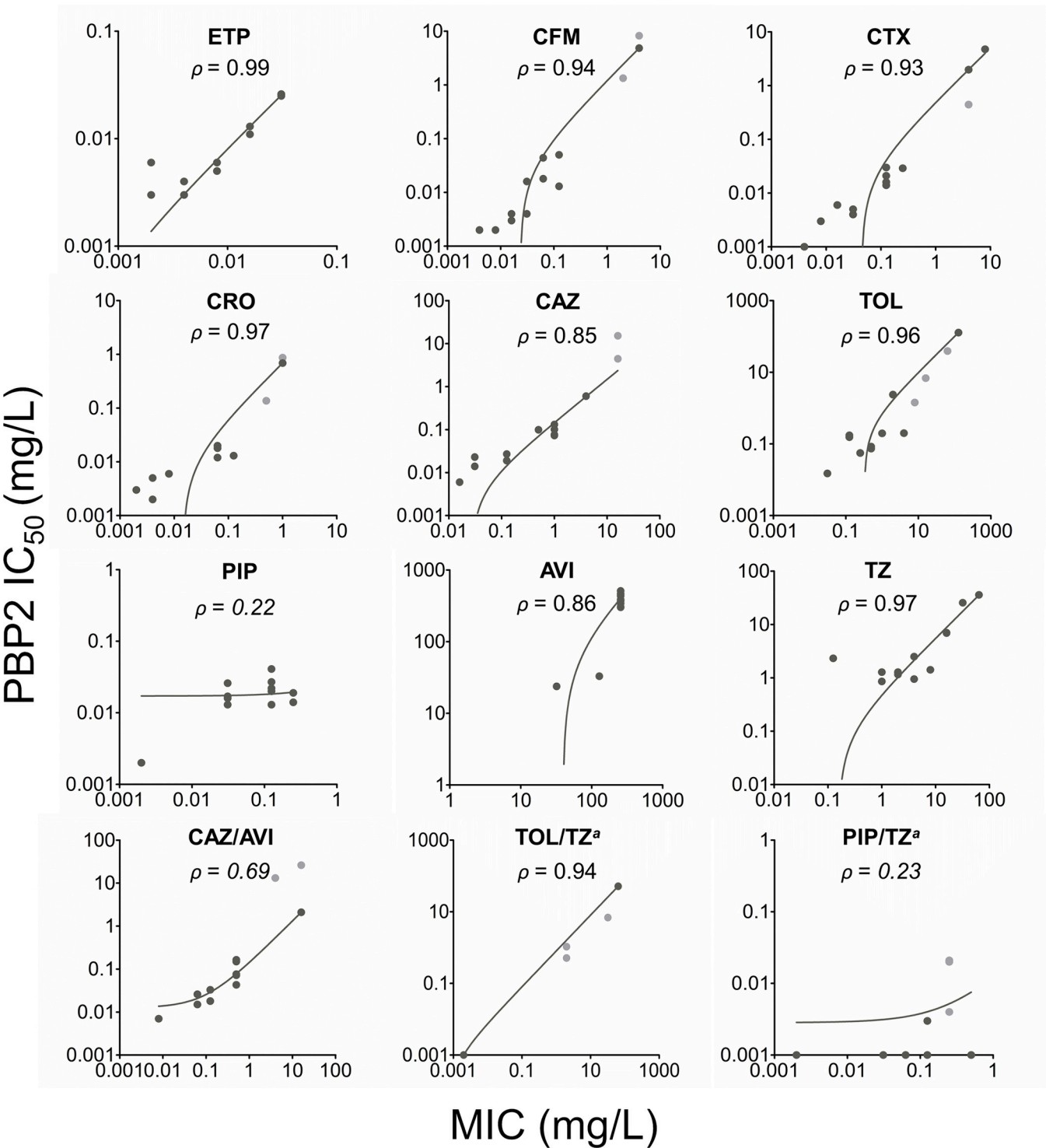

**Fig 3. Scatter plots were employed to illustrate the correlation between PBP2 IC$_{50}$ values and the minimum inhibitory concentrations (MIC) of β-lactam antibiotics, β-lactamase inhibitors (BLI), and the combination of β-lactams with BLI.** Each plot depicts the relationship between PBP2 IC50 and the MICs for each drug across all tested strains. The data were represented on a logarithmic scale and analyzed using a robust fitting method in GraphPad Prism. Light grey dots represent outliers that were included in the analyses. The *N. gonorrhoeae* strains studied were ATCC 19424 and ATCC 49226, as well as clinical strains NG 3, NG 7, NG 12, NG 14, NG 19, NG 20, NG 21, and NG 22, and WHO reference strains NCTC 13820 (WHO X), NCTC 13821 (WHO Y), and NCTC 13822 (WHO Z). Please refer to Table 1 for drug abbreviations. Italicized values indicate low Pearson correlation coefficients ($\rho$). [a] To determine an estimated correlation, all values below 0.002 were rounded to 0.001.

exhibited only minor changes in ertapenem MICs, which varied between 2- and 8-fold (MICs 0.004 to 0.016). In terms of β-lactam/β-lactamase inhibitor combinations, the most significant increase in MICs was observed with ceftazidime/avibactam, which ranged from 4- to 250-fold (MICs 0.064 to 8 mg/L). Changes in ceftolozane/tazobactam MICs were noted exclusively in the NG 12, WHO X, and Y strains, with increases of 2-, 1000-, and 500-fold, respectively; however, no alterations were observed for piperacillin/tazobactam MICs (lower concentration tested 0.016 mg/L) (**Tables 2 and S4**).

## Efflux pumps contribution to antimicrobial resistance

The MIC values of the compounds were evaluated for all isolates in the presence of two efflux pump inhibitors, Carbonyl cyanide 3-chlorophenylhydrazone (CCCP) and Phe-Arg-β-naphthylamide (PAβN), to determine the role of efflux pumps in antimicrobial resistance. Tables 3 and 4 show the MICs of the β-lactams and β-lactam-BLI combinations with the addition of 0.1 mg/L of the EPI CCCP. The comparative analysis indicates that the addition of the inhibitor CCCP resulted in a modest decrease (1- to 4-fold) in the MICs of cefixime, ceftazidime, and ceftazidime/avibactam (**Table 3**). Ceftriaxone exhibited a similar trend, except for strains NG 12 and NG 14, which demonstrated an 8- to 16-fold reduction.

Furthermore, ceftolozane/tazobactam MICs were uniquely reduced in those two strains by 250-fold. It is important to note that the effect of CCCP on ceftolozane/tazobactam may have been underestimated due to the already lower MICs that were below the detection limit (<0.001). No MIC reduction was observed for tazobactam (**S5 Table**), while piperacillin exclusively showed an 8- and 32-fold reduction in the MICs of strains ATCC 49226 and NG 7, respectively. However, the addition of CCCP to piperacillin/tazobactam resulted in a reduction of the MIC by 4- to > 250-fold in all strains, except for ATCC 19424, NG 19, and NG20, which already exhibited piperacillin/tazobactam MICs below 0.002 mg/L. Azithromycin MICs decreased by 4- to 8-fold in all strains except for ATCC 19424, NG 20 and WHO Y (**Table 3**). We also determined the MICs of the two most affected antimicrobials, azithromycin and piperacillin/tazobactam, in the presence of 25 mg/L PAβN (**S5 Table**), and obtained the same results.

We inactivated *mtrD* to confirm that the reduction in the MIC was due to MtrCDE efflux inhibition. Similar to what happens when an efflux inhibitor is introduced; the clinical isolates and WHO ESC-resistant isolates that lacked the MtrCDE pump showed a marked increase in susceptibility to ESCs, penicillin, and azithromycin (**Table 4**). Ceftriaxone, ceftazidime, and piperacillin showed 2.5-fold lower MICs in the *mtrD* inactivated strains, but otherwise the observed values were similar to those obtained with the inhibitors (0.6- to 1.2-fold). Moreover, after *mtrD* inactivation, cefotaxime displayed MICs that were ten times lower.

## Discussion

*Neisseria gonorrhoeae* poses significant therapeutic challenges due to low cure rates, high morbidity, and suboptimal drug-target achievement, even in susceptible strains. Resistance mechanisms including overexpressed efflux pumps, modified porins, β-lactamases, and target site modifications affecting multiple antibiotics create an urgent need for optimized therapies to combat this microorganism [5,9,11,39].

In our previous research, we introduced an extensive dataset that focused on penicillin-binding protein (PBP) occupancy in *N. gonorrhoeae*, particularly for 22 clinically significant β-lactams and β-lactamase inhibitors (BLIs) in two *N. gonorrhoeae* strains, ATCC 19424 and ATCC 49226 [31]. However, there is currently a scarcity of studies that examine the modified affinity of mosaic PBP2 alleles for β-lactams in the presence of all PBP receptors [40,41].

**Table 3. Impact of the efflux pump inhibitor CCCP on the minimum inhibitory concentrations of azithromycin, piperacillin, tazobactam, and piperacillin/tazobactam examined in the studied strains.**

| Strain[a] | CCCP[c] | ETP | ETP + CCCP[e] | FC[d] | CFM | CFM + CCCP[e] | FC[d] | CTX | CTX + CCCP[e] | FC[d] | CRO | CRO + CCCP[e] | FC[d] | AZT | AZT + CCCP[e] | FC[d] | CAZ | CAZ + CCCP[d] | FC[c] | CAZ/AVI[c] | CAZ/AVI[c] + CCCP[d] | FC[c] | TOL/TZ[c] | TOL/TZ[c] + CCCP[d] | FC[c] | PIP | PIP + CCCP[d] | FC[c] | PIP/TZ[c] | PIP/TZ[c] + CCCP[d] | FC[c] |
|---|---|---|---|---|---|---|---|---|---|---|---|---|---|---|---|---|---|---|---|---|---|---|---|---|---|---|---|---|---|---|---|
| ATCC 19424 | <0.5 | 0.004 | 0.004 | 1 | 0.004 | 0.004 | 1 | 0.004 | 0.004 | 1 | 0.002 | 0.002 | 1 | 0.016 | 0.016 | 1 | 0.016 | 0.016 | 1 | 0.008 | 0.008 | 1 | <0.002 | <0.002 | 1 | <0.002 | <0.002 | 1 | <0.002 | <0.002 | 1 |
| ATCC 49226 | 1 | 0.008 | 0.004 | 2 | 0.016 | 0.008 | 2 | 0.008 | 0.008 | 1 | 0.004 | 0.004 | 1 | 0.25 | 0.032 | 8 | 0.032 | 0.016 | 2 | 0.064 | 0.032 | 2 | <0.002 | <0.002 | 1 | 0.125 | 0.016 | 8 | 0.064 | <0.002 | ≥32 |
| NG 3 | 2 | 0.004 | 0.004 | 1 | 0.032 | 0.032 | 1 | 0.125 | 0.125 | 1 | 0.064 | 0.016 | 4 | 0.25 | 0.032 | 8 | 0.5 | 0.25 | 2 | 0.5 | 0.25 | 2 | <0.002 | <0.002 | 1 | 0.064 | 0.032 | 1 | 0.032 | <0.002 | ≥32 |
| NG 7 | 1 | 0.002 | 0.002 | 1 | 0.032 | 0.016 | 2 | 0.016 | 0.008 | 2 | 0.004 | <0.002 | ≥4 | 16 | 4 | 4 | 0.032 | 0.016 | 2 | 0.064 | 0.016 | 4 | <0.002 | <0.002 | 1 | 0.25 | 0.008 | 32 | 0.032 | 0.008 | 4 |
| NG 12 | 1 | 0.032 | 0.016 | 2 | 0.25 | 0.064 | 4 | 0.25 | 0.125 | 2 | 0.125 | 0.016 | 8 | 0.5 | 0.064 | 8 | 1 | 0.5 | 2 | 0.5 | 0.5 | 1 | 2 | 0.008 | 250 | 0.125 | 0.125 | 1 | 0.25 | 0.008 | 32 |
| NG 14 | 2 | 0.032 | 0.016 | 2 | 0.125 | 0.032 | 4 | 0.25 | 0.125 | 2 | 0.125 | 0.008 | 16 | 0.5 | 0.064 | 8 | 1 | 0.5 | 2 | 0.5 | 0.5 | 1 | 2 | 0.008 | 250 | 0.032 | 0.016 | 2 | 0.125 | <0.002 | ≥64 |
| NG 19 | 1 | 0.004 | 0.004 | 1 | 0.008 | 0.008 | 1 | 0.032 | 0.032 | 1 | 0.008 | 0.004 | 2 | 1 | 0.25 | 4 | 0.125 | 0.032 | 4 | 0.125 | 0.064 | 2 | <0.002 | <0.002 | 1 | 0.032 | 0.016 | 2 | <0.002 | <0.002 | 1 |
| NG 20 | 2 | 0.008 | 0.004 | 2 | 0.016 | 0.008 | 2 | 0.032 | 0.016 | 2 | 0.004 | <0.002 | ≥4 | 2048 | 2048 | 1 | 0.125 | 0.032 | 4 | 0.125 | 0.032 | 4 | <0.002 | <0.002 | 1 | 0.125 | 0.125 | 1 | <0.002 | <0.002 | 1 |
| NG 21 | 2 | 0.016 | 0.008 | 2 | 0.064 | 0.032 | 2 | 0.125 | 0.125 | 1 | 0.064 | 0.032 | 2 | 0.25 | 0.032 | 8 | 1 | 0.5 | 2 | 0.5 | 0.5 | 1 | <0.002 | <0.002 | 1 | 0.125 | 0.125 | 1 | 0.125 | <0.002 | ≥32 |
| NG 22 | 1 | 0.008 | 0.004 | 2 | 0.125 | 0.064 | 2 | 0.125 | 0.125 | 1 | 0.064 | 0.032 | 2 | 0.25 | 0.032 | 8 | 1 | 0.5 | 2 | 0.5 | 0.5 | 1 | <0.002 | <0.002 | 1 | 0.25 | 0.125 | 2 | 0.5 | <0.002 | ≥250 |
| WHO X | 2 | 0.032 | 0.016 | 2 | 4 | 4 | 1 | 8 | 8 | 1 | 2 | 2 | 1 | 0.25 | 0.032 | 8 | 16 | 16 | 1 | 16 | 8 | 2 | 64 | 32 | 2 | 0.125 | 0.125 | 1 | 0.25 | 0.063 | 4 |
| WHO Y | 1 | 0.002 | 0.002 | 1 | 4 | 4 | 1 | 4 | 2 | 2 | 1 | 1 | 1 | 0.25 | 0.064 | 4 | 16 | 16 | 1 | 16 | 8 | 2 | <0.002 | <0.002 | 1 | 0.032 | 0.032 | 1 | 0.032 | <0.002 | ≥16 |
| WHO Z | 2 | 0.016 | 0.016 | 1 | 2 | 2 | 1 | 4 | 2 | 2 | 0.5 | 0.5 | 1 | 0.25 | 0.064 | 4 | 4 | 4 | 1 | 4 | 4 | 1 | 32 | 16 | 2 | 0.125 | 0.125 | 2 | 0.25 | 0.064 | 4 |

[a] *N. gonorrhoeae* strains ATCC 19424 and ATCC 49226; clinical strains NG 3, NG 7, NG 12, NG 14, NG 19, NG 20, NG 21 from Hospital Universitario Son Espases (Spain) and NG 22 from Hospital Clínic de Barcelona (Spain); and WHO reference strains NCTC 13820 (WHO X), NCTC 13821 (WHO Y) and NCTC 13822 (WHO Z)

[b] Broth microdilution MICs were performed following CLSI guidelines (38). CCCP, Carbonyl cyanide 3-chlorophenylhydrazone. Please see Table 1 for drug abbreviations

[c] MICs were performed with a fixed concentration of the BLIs tazobactam or avibactam (4 mg/L). The MICs of avibactam or tazobactam were unaffected by the addition of CCCP.

[d] MICs were performed with a fixed concentration of the EPI CCCP (0.1 mg/L)

[e] FC = MIC fold change after the addition of CCCP. Bold numbers indicate a ≥ 4-fold MIC reduction.

**Table 4. Impact of the *mtrD* gene inactivation on the minimum inhibitory concentrations in the studied strains.**

| Strain[a] | MIC of the indicated drug (mg/L)[b] | | | | | | | | | | |
|---|---|---|---|---|---|---|---|---|---|---|---|
| | PenG | ETP | CFM | CTX | CRO | CAZ | CAZ/ AVI[c] | TOL/TZ[c] | PIP | PIP/TZ[c] | AZT |
| ATCC 49226 | 0.5 | 0.008 | 0.016 | 0.008 | 0.004 | 0.032 | 0.064 | <0.002 | 0.125 | 0.064 | 0.25 |
| ATCC 49226 *mtrD*::Km | 0.125 | <0.002 | 0.008 | 0.004 | 0.002 | 0.032 | 0.016 | <0.002 | 0.064 | 0.002 | 0.064 |
| NG 3 | 0.5 | 0.004 | 0.032 | 0.125 | 0.064 | 0.5 | 0.5 | <0.002 | 0.064 | 0.032 | 0.25 |
| NG 3 *mtrD*::Km | 0.032 | 0.004 | 0.008 | 0.004 | 0.008 | 0.125 | 0.064 | <0.002 | 0.008 | <0.002 | 0.032 |
| NG 7 | 0.25 | 0.002 | 0.032 | 0.016 | 0.004 | 0.032 | 0.064 | <0.002 | 0.25 | 0.032 | 16 |
| NG 7 *mtrD*::Km | 0.032 | <0.002 | 0.008 | 0.004 | <0.002 | 0.032 | 0.008 | <0.002 | 0.016 | 0.008 | 8 |
| NG 12 | 1 | 0.032 | 0.25 | 0.25 | 0.125 | 1 | 0.5 | 2 | 0.125 | 0.25 | 0.5 |
| NG 12 *mtrD*::Km | 0.125 | 0.016 | 0.064 | 0.016 | 0.008 | 0.25 | 0.25 | 0.064 | 0.016 | 0.008 | 0.032 |
| NG 14 | 1 | 0.032 | 0.125 | 0.25 | 0.125 | 1 | 0.5 | 2 | 0.032 | 0.125 | 0.5 |
| NG 14 *mtrD*::Km | 0.125 | 0.016 | 0.064 | 0.032 | 0.008 | 0.25 | 0.125 | 0.008 | 0.008 | <0.002 | 0.032 |
| NG 19 | 0.5 | 0.004 | 0.008 | 0.032 | 0.008 | 0.125 | 0.125 | <0.002 | 0.032 | <0.002 | 1 |
| NG 19 *mtrD*::Km | 0.064 | <0.002 | 0.008 | 0.002 | <0.002 | 0.016 | 0.032 | <0.002 | 0.008 | <0.002 | 0.032 |
| NG 20 | 0.5 | 0.008 | 0.016 | 0.032 | 0.004 | 0.125 | 0.125 | <0.002 | 0.125 | <0.002 | 2048 |
| NG 20 *mtrD*::Km | 0.064 | 0.004 | 0.008 | 0.002 | <0.002 | 0.016 | 0.032 | <0.002 | 0.008 | <0.002 | 2048 |
| NG 21 | 1 | 0.016 | 0.064 | 0.125 | 0.064 | 1 | 0.5 | <0.002 | 0.125 | 0.125 | 0.25 |
| NG 21 *mtrD*::Km | 0.064 | 0.002 | 0.008 | 0.004 | 0.004 | 0.125 | 0.125 | <0.002 | 0.008 | <0.002 | 0.016 |
| NG 22 | 1 | 0.008 | 0.125 | 0.125 | 0.064 | 1 | 0.5 | <0.002 | 0.25 | 0.5 | 0.25 |
| NG 22 *mtrD*::Km | 0.064 | 0.004 | 0.016 | 0.008 | 0.008 | 0.125 | 0.125 | <0.002 | 0.016 | <0.002 | 0.032 |
| WHO X | 4 | 0.032 | 4 | 8 | 2 | 16 | 16 | 64 | 0.125 | 0.25 | 0.25 |
| WHO X *mtrD*::Km | 0.5 | 0.016 | 2 | 1 | 0.5 | 4 | 8 | 32 | 0.008 | 0.016 | 0.032 |
| WHO Y | 4 | 0.002 | 4 | 4 | 1 | 16 | 16 | <0.002 | 0.125 | 0.032 | 0.25 |
| WHO Y *mtrD*::Km | 0.064 | <0.002 | 2 | 0.5 | 0.25 | 4 | 4 | <0.002 | 0.004 | <0.002 | 0.032 |
| WHO Z | 2 | 0.016 | 2 | 4 | 0.5 | 4 | 4 | 32 | 0.125 | 0.25 | 0.25 |
| WHO Z *mtrD*::Km | 0.25 | 0.008 | 1 | 0.25 | 0.125 | 1 | 1 | 8 | 0.008 | 0.016 | 0.032 |

[a] Strains were transformed with KH14 strain DNA and with the full-length PCR of the *mtrD*::Km construct leading to *mtrD* inactivation

[b] Agar microdilution MICs were performed following CLSI guidelines [38].

Therefore, the aim of this study was to explore the association of *penA* alleles with PBP occupancy patterns and the reduced outer membrane permeability with the decreased susceptibility to last-line cephalosporins and other potential β-lactam candidates. To achieve this, we determined the PBP-binding IC$_{50}$s of clinical strains and cephalosporin-resistant WHO *N. gonorrhoeae* reference strains carrying different *penA* alleles, including mosaic variants, for the first time.

Analysis showed no significant differences in band intensities of the 3 visible PBPs in reference strains and clinical isolates. Most compounds studied were selective for PBP2, the key target for killing this microorganism [31]. Avibactam and ceftazidime/avibactam were shown to be selective for PBP3, while ertapenem, tazobactam and their combinations simultaneously targeted both PBP2 and PBP3.

The genetic characterization of our selected clinical isolates and WHO reference strains showed that decreased sensitivity to β-lactams, including ESCs, was linked to mutations in *ponA*, *mtrR*, and *porB1b*. However, as previously observed, the presence of a mosaic *penA* allele emerged as the key factor influencing the varying resistance levels observed [5,42,43].

Meanwhile, non-mosaic *penA* alleles II (NG7 and NG 20), V (NG 19) and XXII (ATCC 49226) led to slight but notable increases in $IC_{50}$s for cefixime and ceftriaxone; these effects were even more pronounced for cefotaxime. Additionally, the A501V substitution in isolates NG 3, NG 21 and NG 22, carrying non-mosaic XIII *penA* alleles (MLST ST 7827 and NG-STAR ST 38) resulted in significantly reduced PBP2 affinity and higher MIC values for cephalosporins, piperacillin, and avibactam but not in the isolates carrying *penA* II allele with the same substitution. However, no such effect was observed for tazobactam and ertapenem [44].

The Mosaic *penA* allele XXXIV, which contains the mutations I312M, V316T, D345a, A501V, F504L, N512Y, and G545S, was present in two isolates (NG 12 and NG 14; NG-STAR ST5569) and led to significantly higher PBP $IC_{50}$s for cephalosporins and ertapenem [45]. The resulting MIC values exceeded clinical breakpoints for cefixime and cefotaxime and were only one dilution below for ceftriaxone [3,6,26]. Moreover, the added A501P modification in the *penA* XLII allele (WHO Y) substantially raised the PBP2 $IC_{50}$ values and increased the MICs of antigonococcal compounds well above susceptibility breakpoints [6,39]. The appearance of Mosaic *penA* XLII in strains containing the PBP2 mosaic XXXIV (share the same MLST ST1901) may contribute to the spread of extensively drug-resistant strains globally [5,42]. The replacement of valine with proline at position A501 leads to considerable increases in cephalosporin PBP2 $IC_{50}$s due to significant structural changes; however, this substitution had a lesser impact on other β-lactams such as penicillins, ertapenem, and tazobactam, the latter retaining wild type levels [6,37,44]https://www.ncbi.nlm.nih.gov/pmc/articles/PMC3294892/.

The *penA* XXXVII mosaic allele (WHO X) produces a PBP2 variant with specific alterations including A311V, V316P, and T483S. There is no change at position 501 in comparison to allele XXXIV [5]. These alterations were responsible for maximally increased cephalosporins $IC_{50}$s and resulted in the highest increase in the MICs of all β-lactams tested. This *penA* allele has demonstrated the lowest rates of acylation of PBP2 by up to 12,000-fold when compared to the wild type variants [46]. Furthermore, this variant maintained resistance to penicillin and showed reduced susceptibility to ertapenem, with the highest $IC_{50}$ and MIC values among the strains studied [5,37].

The variant of *penA* LXIV (WHO Z), with two out of three key changes (A311V and T483S, but not V316P) had the same MLST ST 7363 as the XXXVII variant [7,37]. The lack of the V316P alteration led to slightly decreased values for cephalosporins $IC_{50}$s and MICs above susceptibility breakpoints and a somewhat smaller effect on PBP2 affinities to penicillins and ertapenem.

The transformation experiments, as previously outlined, provided further evidence suggesting that the different *penA* alleles, particularly the mosaic variants, were responsible for the increased resistance to ESCs. Nevertheless, the minimum inhibitory concentrations of the ESCs in the recipient strain vary due to the lack of additional resistance determinants (e.g. *mtrR*, *penB*, etc.) [5,6].

The correlation between *penA*, *ponA*, *mtrR*, and *porB1b* in relation to β-lactam resistance has been widely studied [3–6,24,26,42,46–49]. In our study, a more pronounced synergistic effect was observed for ceftriaxone compared to cefixime, and an even greater effect for the no longer recommended cefotaxime [37,47]. Among third generation cephalosporins, ceftazidime was particularly affected despite its slightly lower acylation rates compared to ceftriaxone [30].

The effectiveness of β-lactams in *N. gonorrhoeae* is influenced not only by their PBP2 affinity but also acylation rate constants, permeation through porins, susceptibility to efflux, and their interaction with the *ponA* transpeptidase (PBP1) [30]. The optimization of recently developed cephalosporins has focused on antimicrobial activity rather than target inhibition, resulting in improved Gram-negative spectrum by enhancing periplasmic accumulation. Such feature poorly correlates with target inhibition [30,50]. Fifth-generation ceftolozane exhibits

very poor acylation of PBP2 and was found to be the least effective cephalosporin (higher $IC_{50}$s) while being most impacted by resistance determinants [30,50,51].

The addition of 4 mg/L of tazobactam to ceftolozane resulted in a significant decrease in PBP2 $IC_{50}$s across all strains except for those with mosaic *penA* alleles XXXIV (NG 12 and NG 14), XXXVII (WHO X) and LXIV (WHO Z), which exhibited tazobactam PBP2 $IC_{50}$s close to or above 4 mg/L and MICs ranging from 16 to 64 mg/L. The MICs showed consistent reductions ranging from 32 to 4000-fold. Strain WHO Y, carrying the XLII mosaic *penA* allele, demonstrated the greatest reduction in $IC_{50}$s and MICs (128,000-fold) among the strains with mosaic *penA* alleles. As previously stated, it is worth noting that this particular *penA* variant is known for its unexpectedly low MICs of penicillins and carbapenems [6,37]. The change from Ala501 to proline may have had a significant impact on altering secondary structure leading to lower tazobactam PBP2 $IC_{50}$ values similar to wild-type strains lacking any resistance determinants.

The addition of tazobactam to piperacillin resulted in a broader and significant decrease in PBP2 $IC_{50}$s, but the observed reductions in MICs were unexpectedly less prominent and only seen in specific strains. Additionally, the combination led to higher MICs for strains with a tazobactam MIC above 4 mg/L. It is challenging to explain this difference solely based on variations in acylation rate constants between ceftolozane and piperacillin [30]. The possibility that both compounds compete for entrance or differences in target site penetration influenced by efflux seems more plausible.

The antigonococcal cephalosporins MICs reduction following CCCP addition was generally minimal, except for strains harboring mosaic *penA* alleles II, XIII and mosaic XXXIV. Previous studies have demonstrated that inactivating the *mtrCDE* pump, above all other efflux systems, can reduce MICs of cephalosporins [20,52]. When we inactivated *mtrD* in our strains we observed a higher MIC reduction, which in the case of the cephalosporin resistant isolates, was comparable to that obtained previously [20].

CCCP disrupts oxidative phosphorylation and the proton gradient in bacterial membranes, impacting cell envelope components such as porins and efflux pumps. Inactivation of these pumps may lead to varied outcomes due to enhanced cell envelope permeability to antimicrobials [20,26]. CCCP addition to tazobactam or piperacillin, led to a significant decrease in MICs for the latter in two of the strains under study. However, when combined with piperacillin/tazobactam, a noticeable reduction in MIC was seen across all tested strains irrespective of the presence of a mosaic *penA* allele. The same outcome was observed upon *mtrD* inactivation. This pattern was evident in all strains except those that had already achieved minimal MIC values (< 0.002/4 mg/L) for the piperacillin/tazobactam combination [52].

Ceftolozane/tazobactam MICs upon EPI addition or *mtrD* inactivation showed a significant decrease in strains NG 12 and NG 14 with the *penA* XXXIV variant. Strains carrying *penA* alleles XXXVII and LXII experienced a minor reduction (2-fold). However, tazobactam impact on the combination may have been underestimated due to its ability to lower MICs to levels below the lowest concentration tested (0.002/4 mg/L) when added to ceftolozane.

The enhanced *mtrD* inactivation effect that occurs when ceftolozane or piperacillin is combined with tazobactam may support a synergistic effect on efflux pump activity when both medications are present in the periplasm or a competitive interaction between the two compounds for entry through PorB1b mutated porin. Since a mutation in the efflux regulator *mtrR* is required to develop high-level resistance to penicillins and cephalosporins, which is imparted by porin porB1b variants (*penB*), the effect may be a combination of both events [14,15]. The observed impact may result from both occurrences as a mutation in *mtrR* is required to develop high-level resistance to penicillins and cephalosporins mediated by porin *porB1b* (*penB*).

The MtrCDE system found in *N. gonorrhoeae* belongs to the hydrophobic and amphiphilic efflux resistance-nodulation-division family of efflux pumps, alongside similar systems such as *Escherichia coli* and *Klebsiella pneumoniae* AcrAB-TolC, as well as *Pseudomonas aeruginosa* MexAB-OprM [53]. Piperacillin, ceftolozane, and tazobactam are likely substrates of this pump in those species [51]. Knocking out the *acrB* and the *oprM* genes decreases the MIC of piperacillin/tazobactam, and ceftolozane/tazobactam suggesting tazobactam expulsion by the respective pumps. Surprisingly, efflux pump inactivation had limited effects on individual drugs compared to their combination in both bacterial species [54–57].

Since azithromycin has been commonly utilized in combination with ceftriaxone, we examined the impact of the efflux impairment on our strains, comprising one isolate exhibiting low-level resistance to azithromycin (due to 23S rRNA C2611T mutation) and another showing high-level resistance (owing to 23S rRNA A2059G mutation). In all sensitive strains or those with low-level resistance, susceptibility to azithromycin was increased 4- to 16-fold. However, in the highly resistant strain, it had no effect. Our experiments consistently showed changes in permeability within the cell envelope, which presumably affected PorB1b and MtrCDE, while no mutations were found in PilQ or the promoters of MacA-MacB and NorM pumps [3,5,20].

Ertapenem did not exhibit a significant advantage over ceftriaxone for *N. gonorrhoeae* strains with lower ceftriaxone MICs, as shown in previous studies from Unemo et al. and Livermore et al. [43,58]. However, it displayed low MICs for all strains resistant to ceftriaxone and remained highly effective against isolates with significant clinical resistance to azithromycin. Additionally, it showed the lowest $IC_{50}$ values for PBP2, which correlated entirely with the MICs, suggesting no impact from efflux or porins—a finding supported by its combination with EPIs and *mtrD* inactivation. Furthermore, ertapenem is not affected by plasmid-mediated β-lactamases such as $bla_{TEM}$ [58].

On the other hand, the observed piperacillin MICs, the second most effective compound, were 6-fold lower than their PBP2 $IC_{50}$s on average. The newly published research from Turner et al. revealed that piperacillin acylates PBP2 at a rate 12 times higher than cefoperazone and 85 times higher than ceftriaxone, with a lower MIC against H041 (WHO X) compared to ceftriaxone. Interestingly, ureidopenicillins have minimal affinity for PBP2, suggesting their inhibitory potency is attributed to a faster acylation step in the reaction when compared to cephalosporins [59]. In addition, piperacillin exhibits a higher level of resistance to degradation by specific plasmid-mediated β-lactamases in comparison to other non-methoxy penicillins [60]. Moreover, the addition of tazobactam improved the effectiveness of piperacillin and has the potential to decrease MICs in strains expressing β-lactamases (even ESBLs) [60]. Furthermore, inhibition of efflux by CCCP and PaβN or *mtrD* inactivation enhanced the effectiveness of this combination against strains exhibiting *mtrR* alterations and mosaic *penA* alleles. It is noteworthy that the decoy target PBP3 seemed to be fully saturated, which was also evident in the case of ertapenem [28].

Piperacillin, in combination with tazobactam, as well as ertapenem, may be regarded as a potential therapeutic alternative for gonorrhea (where accessible), especially in instances of ESC-resistant strains, notwithstanding their brief half-lives and the requirement for parenteral administration [6,61].

This study's findings will enhance our understanding of how PBP2 variants and enhanced efflux relate to susceptibility to the last-line, future, and alternative treatments for gonorrhea infections. Our findings shed light on the selective nature of various compounds for the different PBPs, allelic variants and permeability, highlighting the intricate nature of antimicrobial resistance in this microorganism. Our research complements the kinetic binding studies and crystallization of individual PBPs, offering valuable insights for extending the efficacy of

antimicrobials, improving dosage regimens for current antimicrobial agents, and advancing the development or reevaluation of new antimicrobial drugs and gonococcal EPIs. Overall, our findings endorse the potential of ertapenem and piperacillin alone or in combination with tazobactam as therapies for complicated gonorrhea and establish a basis for developing β-lactams with enhanced efficacy against ESC-resistant *N. gonorrhoeae*. However, additional research in this field is essential to comprehend the specific ways in which EPIs impact drug sensitivity and to inform the development of adjuvants that could be therapeutically effective in cephalosporin resistant *N. gonorrhoeae* infections.

## Materials and methods

### Bacterial strains and *in vitro* susceptibility testing

The bacterial strains and isolates utilized in this study included *N. gonorrhoeae* clinical isolates NG 3, NG 7, NG 12, NG 14, NG 19, NG 20, and NG 21 collected between the years 2016–2023 from Hospital Universitario Son Espases, along with isolate NG 22 from Hospital Clínic de Barcelona (multicenter study). We selected the strains based on varying degrees of susceptibility to β-lactams, specifically excluding isolates resistant to penicillin. Our goal was to investigate isolates with different *penA* alleles rather than those producing $bla_{TEM}$. The MICs database from isolates belonging to our hospital and from a multicenter study was grouped and rank ordered according to their sensitivity to ceftriaxone, cefixime and cefotaxime. We subsequently chose eight isolates displaying varied sensitivity to the three cephalosporins, with MICs spanning from 0.004 to 0.250 mg/L. The lower (0.002–0.004 mg/L) and upper (0.5–8 mg/L) sensitivity ranges were completed with the wild type collection strains ATCC 19424 and ATCC 49226 and WHO reference strains (WHO X—NCTC 13820, WHO Y—NCTC 13821, and WHO Z—NCTC 13822) belonging to H41, F89, and A8806 strains with "high-level resistance" to extended-spectrum cephalosporins respectively [37].

For the present study, isolates were plated on GC agar (bioMérieux) and their species identification was confirmed using the MALDI-TOF MS (Bruker Daltonik). The minimum inhibitory concentrations (MICs) of penicillin, ertapenem, cefixime, cefotaxime, ceftriaxone, ceftazidime, ceftolozane, piperacillin, avibactam, tazobactam, ceftazidime/avibactam, ceftolozane/tazobactam, piperacillin/tazobactam, azithromycin, tetracycline and ciprofloxacin were determined by the ETEST method (bioMérieux, Solna, Sweden), according to the manufacturer's instructions, on BBL Chocolate II Agar (GC II Agar with Haemoglobin and IsoVitaleX) agar plates, and by the standard Clinical and Laboratory Standards Institute (CLSI) agar diffusion and broth microdilution methods [62]. MICs with sub-MIC concentrations of the efflux pump inhibitors (EPIs) carbonyl cyanide 3-chlorophenylhydrazone (CCCP) at a concentration of 0.1 mg/L, were determined for all compounds. MICs were also evaluated with the EPI Phe-Arg-β-naphthylamide (PAβN) at a concentration of 25 mg/L to validate the results in selected drugs. The resulting fold changes in MIC after the addition of CCCP were calculated as the ratio between the antibiotic's MIC level without CCCP and that with CCCP added. A significant impact of the efflux pump inhibitor in isolates was defined as a $\geq$ 4-fold reduction in antibiotic MIC following CCCP addition.

Ertapenem was purchased from Merck Sharp & Dohme (Haarlem, Netherlands); penicillin G from Laboratorio Reig Jofré SA (Barcelona, Spain); cefotaxime, ceftriaxone and ceftazidime from Laboratorios Normon (Madrid, Spain), piperacillin cefixime, ceftolozane, avibactam, tazobactam, ciprofloxacin and tetracycline from MedChem Express (Sollentuna, Sweden). MIC values **were determined from at least three independent experiments.** Where breakpoints were available, the susceptible (S), and resistant (R) categorization was based on the interpretative criteria from EUCAST (www.eucast.org). The CLSI susceptible-only interpretive

breakpoint ($\leq 1$ mg/L) for azithromycin was utilized (which also serves as the epidemiological cutoff value, marking the end of the wild-type susceptibility distribution). High-level resistance is identified when the MIC for azithromycin exceeds >16 mg/L [62]. Each determination was performed at least three times using new bacterial suspensions on separate batches of agar plates and the consensus MIC was reported.

## PBP-binding assays

The $IC_{50}$ binding affinities of PBPs were assessed using previously described protocols [32]. Briefly, *N. gonorrhoeae* cultures were grown in phosphate-buffered gonococcal medium (GCP) with 1/100 volume 4.2% sodium bicarbonate and 1/100 volume Kellogg's supplement at 37˚C in a shaking incubator (180 rpm). Mid-exponential growing cultures (6.5 log10 CFU/mL) were collected. The cells underwent centrifugation, washing, and resuspension in a solution containing $KH_2PO_4$ with NaCl pH7.5; they were then sonicated and subjected to ultracentrifugation (150,000 $g$) to collect bacterial membranes. The membrane preparations were treated with Bocillin FL for visualization on SDS-PAGE using a biomolecular imager followed by quantification through ImageQuantTL 8.1 (GE Healthcare Bio-Sciences AB, Björkgatan, 30 751 84 Uppsala). When no measurable binding was observed for the primary PBP target, an extended concentration range was used as necessary (0.001–0.125 or 2–512 mg/L when indicated). The binding affinities were expressed as β-lactam or BLI concentrations that inhibited half-maximally the Bocillin FL binding ($IC_{50}$s), which was determined from three independent experiments.

## Transformation assays

In order to establish that the distinct *penA* allele present in our isolates was responsible for the heightened resistance to β-lactams, we PCR amplified the full-length *penA* from isolates NG 3, NG 12, NG 14, NG 21, NG 22, WHO X, WHO Y, and WHO Z, subsequently transforming it into the wild type strain FA 1090, as outlined in previous studies [5,23]. The complete *penA* allele was sequenced in all resulting transformants.

In order to assess the specific contribution of the *mtrCDE* efflux pump to the observed resistance we constructed pUCP*mtrD*::Km from FA 1090 chromosomal DNA using previously reported methods [20,63]. The kanamycin (Km) resistance cassette was sourced through PCR amplification of a plasmid aqcuired from the gene synthesis services provided by Genscript Inc. [64]. DNA from the previously constructed *N. gonorrhoeae* KH14 strain (FA19 *mtrD*::Km) was kindly provided By Professor William Shafer [63]. The pUCP*mtrD*::Km digested product and the KH14 DNA were used to transform isolates ATCC 49226, NG 3, NG 12, NG 14, NG 21, NG 22, WHO X, WHO Y, and WHO Z to $Km^R$ (50 mg/L). The *mtrD* allele was amplified using previously described primers in all resulting transformants to verify insertion of the $Km^R$ cassette in the *mtrD* gene [63].

## Whole genome sequencing

Total DNA was isolated using a commercial capture system (High Pure PCR Template Preparation Kit, Roche Diagnostics) and indexed paired-end libraries were generated using the Illumina DNA Prep library preparation kit (Illumina Inc, USA) and then sequenced on an Illumina MiSeq platform. The reads for each isolate were mapped against the genome of the *N. gonorrhoeae* reference strain ATTC 19424 (NCTC 8375) using Bowtie 2 software, version 2.2.6 (http://bowtie-bio.sourceforge.net/bowtie2/index.shtml) [65]. Pileups and raw files of the mapped reads were obtained by using SAMtools, version 0.1.16 (https://sourceforge.net/projects/samtools/files/samtools/) [66], and PicardTools, version 1.140 (https://github.com/

broadinstitute/picard). Read alignments surrounding all putative indels were realigned using the Genome Analysis Toolkit (GATK), version 3.4–46 (https://www.broadinstitute.org/gatk/) [67]. The list of SNPs was compiled from the raw files that met the following criteria: a quality score of >50, a root mean square (RMS) mapping quality of >25 and a coverage depth of >3. Indels were extracted from the total pileup files by use of the following criteria: a quality score of >250, an RMS mapping quality of >25 and a coverage depth of >3. SNPs and indels for each isolate were annotated by using SnpEff software version 4.3 (http://snpeff.sourceforge.net/index.html) [68].

## Data analysis

The assignment of allele numbers and sequence types was carried out by referencing the *N. gonorrhoeae* multiantigen sequence typing (NG-MAST) (http://www.ng-mast.net), multilocus sequence typing (MLST) (https://pubmlst.org/neisseria), and *N. gonorrhoeae* sequence typing for antimicrobial resistance (NG-STAR) (https://ngstar.canada.ca). Furthermore, verification of antimicrobial resistance determinants was performed using PubMLST (https://pubmlst.org/) and ResFinder (https://cge.cbs.dtu.dk/services/ResFinder/).

GraphPad Prism version 9.5.1 (GraphPad Software, Boston, Massachusetts USA) was used for graphical representation and statistical analysis. Quantitative variables were compared using paired or unpaired Student's *t* test (one or two independent groups) or one-way ANOVA with post hoc Tukey's multiple comparison test (multiple independent groups). To assess the correlation between MIC values and PBP2 $IC_{50}$, the data was plotted on a logarithmic scale and fitted with a robust fit. Pearson correlation coefficients ($\rho$) are shown in the figure.

## Supporting information

**S1 Table. Genetic characteristics of *Neisseria gonorrhoeae* strains ATCC 19424 and 49226, clinical strains and WHO reference strains X, Y and Z.** [a] *N. gonorrhoeae s*trains ATCC 19424 and ATCC 49226; clinical strains NG 3, NG 7, NG 12, NG 14, NG 19, NG 20, NG 21 from Hospital Universitario Son Espases (Spain) and NG 22 from Hospital Clínic de Barcelona (Spain); and WHO reference strains NCTC 13820 (WHO X), NCTC 13821 (WHO Y) and NCTC 13822 (WHO Z). MLST, multilocus sequence typing; NG-MAST, *Neisseria gonorrhoeae* multiantigen sequence typing; NG-STAR, *Neisseria gonorrhoeae* Sequence Typing for Antimicrobial Resistance; ST, sequence type; WT, wild type. New NG-STAR profiles ST5411 and ST5569 were described. [b] The sequences and complete genomes for the *N. gonorrhoeae* ATCC strains 19424 and 49226, were obtained from the ATCC (American Type Culture Collection) genome portal. [c] The sequences for the *N. gonorrhoeae* clinical strains, were obtained from whole genome sequencing (WGS). [d] The sequences and complete genomes for the *N. gonorrhoeae* NCTC strains 13820 (WHO X), NCTC 13821 (WHO Y) and NCTC 13822 (WHO Z), were obtained from the BioProject PRJEB14020.
(PDF)

**S2 Table. PBP $IC_{50}$ ± SD of β-lactam antibiotics and BLIs in *N. gonorrhoeae* strains.** [a] *N. gonorrhoeae s*trains ATCC 19424 and ATCC 49226; clinical strains NG 3, NG 7, NG 12, NG 14, NG 19, NG 20, NG 21 from Hospital Universitario Son Espases (Spain) and NG 22 from Hospital Clínic de Barcelona (Spain); and WHO reference strains NCTC 13820 (WHO X), NCTC 13821 (WHO Y) and NCTC 13822 (WHO Z). [b] PBP, penicillin-binding proteins. [c] This table shows the concentration of β-lactam required to inhibit 50% of Bocillin FL compared to a control with no drug. The mean values from three experiments are presented. The

abbreviations used are as follows: ETP for ertapenem, CFM for cefixime, CTX for cefotaxime, CRO for ceftriaxone, CAZ for ceftazidime, TOL for ceftolozane, PIP for piperacillin, AVI for avibactam, TZ for tazobactam, CAZ/AVI for ceftazidime/avibactam, TOL/TZ for ceftolozane/tazobactam, and PIP/TZ for piperacillin/tazobactam. [d] When the primary PBP target was not inhibited by the regular concentrations (0.001 to 0.125 mg/L or 0.016 to 2 mg/L), an extended range of 1 to 512 mg/L was used. For CAZ/AVI, TOL/TZ, and PIP/TZ, a fixed concentration of the BLIs avibactam or tazobactam at 4 mg/L was used.
(PDF)

**S3 Table. MIC and PBP2 IC$_{50}$ ratio.** [a] Pearson correlation coefficient. [b] To calculate an estimated ratio, all MIC values less than 0.002 were converted to 0.001.
(PDF)

**S4 Table. Ratio between FA 1090 MIC and *penA* FA 1090 transformants.** [a] The following *N. gonorrhoeae* strains were studied: ATCC 19424 and ATCC 49226; clinical strains NG 3, NG 7, NG 12, NG 14, NG 19, NG 20, NG 21 from Hospital Universitario Son Espases (Spain) and NG 22 from Hospital Clínic de Barcelona (Spain); and WHO reference strains NCTC 13820 (WHO X), NCTC 13821 (WHO Y) and NCTC 13822 (WHO Z). [b] Broth microdilution MICs were performed following CLSI guidelines [1]. The antibiotics/EPI tested were ertapenem (ETP), cefixime (CFM), cefotaxime (CTX), ceftriaxone (CRO), ceftazidime (CAZ), ceftolozane (TOL), piperacillin (PIP), avibactam (AVI), tazobactam (TZ), ceftazidime/avibactam (CAZ/AVI), ceftolozane/tazobactam (TOL/TZ), piperacillin/tazobactam (PIP/TZ), azithromycin (AZT) carbonyl cyanide 3-chlorophenylhydrazone (CCCP) and PAβN (Phe-Arg-β-naphthylamide). [c] PIP/TZ, TOL/TZ and CAZ/AVI MICs were conducted using a fixed concentration of 4 mg/L BLI avibactam or tazobactam. [c] MICs were performed with a fixed concentration of the EPI CCCP (0.1 mg/L) and PAβN (25 mg/L). [d] FC: MIC fold change after the addition of PaβN or CCCP. Bold numbers indicate a ≥4-fold MIC reduction when combined with EPI.
(PDF)

**S5 Table. Impact of the efflux pump inhibitors PAβN and CCCP on the minimum inhibitory concentrations of azithromycin, piperacillin, tazobactam, and piperacillin/tazobactam examined in the studied strains.** a The following N. gonorrhoeae strains were studied: ATCC 19424 and ATCC 49226; clinical strains NG 3, NG 7, NG 12, NG 14, NG 19, NG 20, NG 21 from Hospital Universitario Son Espases (Spain) and NG 22 from Hospital Clínic de Barcelona (Spain); and WHO reference strains NCTC 13820 (WHO X), NCTC 13821 (WHO Y) and NCTC 13822 (WHO Z). b Broth microdilution MICs were performed following CLSI guidelines [1]. The antibiotics/EPI tested were ertapenem (ETP), cefixime (CFM), cefotaxime (CTX), ceftriaxone (CRO), ceftazidime (CAZ), ceftolozane (TOL), piperacillin (PIP), avibactam (AVI), tazobactam (TZ), ceftazidime/avibactam (CAZ/AVI), ceftolozane/tazobactam (TOL/TZ), piperacillin/tazobactam (PIP/TZ), azithromycin (AZT) carbonyl cyanide 3-chlorophenylhydrazone (CCCP) and PAβN (Phe-Arg-β naphthylamide). c PIP/TZ, TOL/TZ and CAZ/AVI MICs were conducted using a fixed concentration of 4 mg/L BLI avibactam or tazobactam. c MICs were performed with a fixed concentration of the EPI CCCP (0.1 mg/L) and PAβN (25 mg/L). d FC: MIC fold change after the addition of PaβN or CCCP. Bold numbers indicate a ≥4-fold MIC reduction when combined with EPI.
(PDF)

**S1 Fig. Penicillin-binding protein occupancy dataset for 7 β-lactams, 2 β-lactamase inhibitors and combinations of 3 β-lactams/β-lactamase inhibitors in *Neisseria gonorrhoeae* strains ATCC 19424, ATCC 49226; clinical strains NG 3, NG 7, NG 12, NG 14, NG 19, NG 20, NG 21 and NG 22; WHO reference strains X, Y and WHO Z, at a global analyzed**

**concentration range of 0.001 to 512 mg/L.** *CFM, 0.25–32 mg/L; CTX, 0.125–16 mg/L; CRO, 0.016–2 mg/L; CAZ, 0.25–32 mg/L; TOL, 1–128 mg/L; PIP, 0.016–2 mg/L; AVI, 2–256 mg/L; TZ, 1–128 mg/L; CAZ/AVI, 0.25–32 mg/L (avibactam 4 mg/L fixed concentration); TOL/TZ, 1–128 mg/L (tazobactam 4 mg/L fixed concentration). PBP-binding assay gels for N. gonorrhoeae strains ATCC 19424, ATCC 49226; clinical strains NG 3, NG 7, NG 12, NG 14, NG 19, NG 20, NG 21 and NG 22; WHO reference strains NCTC 13820 (WHO X), NCTC 13821 (WHO Y) and NCTC 13822 (WHO Z). 1 Drugs tested were ertapenem (ETP), cefixime (CFM), cefotaxime (CTX), ceftriaxone (CRO), ceftazidime (CAZ), ceftolozane (TOL), piperacillin (PIP), avibactam (AVI), tazobactam (TZ), ceftazidime/avibactam (CAZ/AVI), ceftolozane/tazobactam (TOL/TZ), and piperacillin/tazobactam (PIP/TZ). The antibiotic-bound PBP-containing membrane preparations were label with 25 µM Bocillin FLTM. Labeled PBPs were separated by SDS-PAGE and detected using a fluorimager. The global range of concentrations tested was 0.001 to 512 mg/L.
(PDF)

## Acknowledgments

We extend our sincere gratitude to Professor William Shafer and Jacqueline Balthazar from Emory University for providing us with DNA from the previously constructed *N. gonorrhoeae* KH14 strain (FA19 *mtrD*::Km). We express our gratitude to Cristina Pitart of Hospital Clínic de Barcelona for providing us with the NG 22 *N. gonorrhoeae* isolate.

## Author Contributions

**Conceptualization:** Bartolome Moya.

**Data curation:** Silvia López-Argüello, Antonio Oliver, Bartolome Moya.

**Formal analysis:** Silvia López-Argüello, Gabriel Cabot, Antonio Oliver, Bartolome Moya.

**Funding acquisition:** Bartolome Moya.

**Investigation:** Silvia López-Argüello, Eva Alcoceba, Paula Ordóñez, Biel Taltavull, Gabriel Cabot, Maria Antonia Gomis-Font, Bartolome Moya.

**Methodology:** Silvia López-Argüello, Eva Alcoceba, Paula Ordóñez, Antonio Oliver, Bartolome Moya.

**Project administration:** Bartolome Moya.

**Resources:** Antonio Oliver, Bartolome Moya.

**Software:** Paula Ordóñez, Biel Taltavull, Gabriel Cabot, Maria Antonia Gomis-Font, Bartolome Moya.

**Supervision:** Silvia López-Argüello, Antonio Oliver, Bartolome Moya.

**Validation:** Silvia López-Argüello, Antonio Oliver, Bartolome Moya.

**Visualization:** Gabriel Cabot, Antonio Oliver, Bartolome Moya.

**Writing – original draft:** Silvia López-Argüello, Eva Alcoceba, Paula Ordóñez, Biel Taltavull, Gabriel Cabot, Maria Antonia Gomis-Font, Antonio Oliver, Bartolome Moya.

**Writing – review & editing:** Silvia López-Argüello, Eva Alcoceba, Antonio Oliver, Bartolome Moya.

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
