## [Decision Letter · Decision Letter 0]

12 Jun 2024

Dear Dr Moya,

Thank you very much for submitting your manuscript "Differential contribution of PBP occupancy and efflux on the effectiveness of β-lactams at their target site in clinical isolates of Neisseria gonorrhoeae." for consideration at PLOS Pathogens. Your manuscript was reviewed by members of the editorial board and by two independent reviewers. In light of the reviews (below), we would like to invite the resubmission of a revised version that takes into account the reviewers' comments.

Please respond to all of the comments from both reviewers. In particular, you need to verify your experiments with the efflux pump inhibitors as indicated by Reviewer 1; specifically, you should use an mtrD null mutant in order to compare your results with those of previous studies. In addition, as noted by Reviewer 2, either evaluate other isolates carrying a PBP2 allele or clone it into FA1090.

We cannot make any decision about publication until we have seen the revised manuscript and your response to the reviewers' comments. Your revised manuscript may be sent to reviewers for further evaluation.

Thank you again for your submission. We hope that our editorial process has been constructive so far, and we welcome your feedback at any time. Please do not hesitate to contact us if you have any questions or comments.

Sincerely,

Jose Luis Balcazar, Ph.D. 

Academic Editor

PLOS Pathogens

D. Scott Samuels

Section Editor

PLOS Pathogens

Michael Malim

Editor-in-Chief

PLOS Pathogens

orcid.org/0000-0002-7699-2064

Reviewer's Responses to Questions

**Part I - Summary**

Reviewer #1: This paper rigorously shows the impact of certain beta-lactams on the major PBPs of Neisseria gonorrhoeae, the influence of a multi-drug efflux pump system on resistance to multiple antibiotics in this class as well as the efficacy of potential combination drugs. The work was carefully performed, and the data seem solid. A minor weaknesses is the lack of establishing that the efflux pump inhibitors that were used can produce a phenotype similar to what is observed with a efflux pump null nutant.

Reviewer #2: This study investigates the relationship between PBP alleles and susceptibility to antibiotics used to treat gonorrhoeae, including last line cephalosporins. By carrying out multiple MIC assays with different drug combination, the authors identified the importance of PBP alleles in resistance against beta lactams and cephalosporin, and showed that inhibiting efflux pumps decreased the MICs of these antibiotics approx 10-fold (or less), which is in line with previous studies.

Additionally, the authors have done a lot of work isolating PBPs to identify relationships between the biochemical relationship between PBP alleles and their affinities with different drugs and drug combinations, with the MICs of different antibiotics. There have been previous related studies establishing the link between different PBP alleles and beta lactam derivatives; despite that, this study stands out in two ways: firstly, it provides biochemical support with PBPs and the corresponding drug targets, rather than relying only on a correlation between PBP alleles observed and MIC values observed. Secondly, the study focuses on evaluating the effectiveness of the few remaining antibiotics that can be used to treat gonorrhoeae and is therefore potentially clinically important.

**Part II – Major Issues: Key Experiments Required for Acceptance**

Reviewer #1: The work would be considerably strengthened if the authors used an mtrD null mutant instead onfefflux pump inhibitors in their work. Without this, it is not really possible to correlate their findings with that of Golparian et al, which they reference. Furthermore, the authors should include penicillin in their work on the impact of the MtrCDE efflux pump. If they can get the same result as as Golparian et al obtained with strain H041 vs H041 mtrD::kan then i would be satisfied.

One comment that is of interest: ertapenem was used to treat the patient in England with an MDR gonorrheal infection in 2016. This is an expensive antibiotic and likely cost-prohibitive for low-middle income countries. In addition, IV treatment required extensive clinical management.

Reviewer #2: The authors established a link between PBP IC50 and MICs using only eight strains; however, these PBP alleles were never cloned into a standard strain to determine their importance in resistance of the antibiotic agents tested. It would be highly beneficial to either 1) investigate other isolates with PBP2 alleles identical to those which were tested and determine what these isolates' MICs of the antibiotics tested are to further validate the conclusions on a global population scale, or 2) clone these into eg. FA1090 and measure changes in MIC to confirm that the PBP alleles described are indeed responsible for changes in MICs (with reference to Figs 2 and 3)

**Part III – Minor Issues: Editorial and Data Presentation Modifications**

Reviewer #1: Their are a number of minor editorial issues that need to be corrected.

1. Line 63: more recent estimates indicate that there is ca. 82 million cases of gonorrhea worldwide/year.

2. Line 100:The mtrR and mtrCDE promoters are not shared but overlap on different strands at the -35 hexamer. The paper by Hagman et al, 1995 should be referenced.

3. line 119: insert the word "above" after "consistently"

4. line 185: insert the word "of" after "any"

5. line 328: there seems to be a word missing after: ceftriaxone and ....

6. sone references are incomplete-see #57 as an example

7. Figure 3 is very confusing as the legend indicates that multiple strains were studied but the figure seems to be the result of 1 strain with multiple antibiotics.

Reviewer #2: -Please italicise gene names (eg penA in line 97)

-Line 243: "Said isolates" - unclear which isolates these said isolates referred to. I suppose these are the resistant ones?

-Line 271: "This is led" - no need for is

-Consider including the Chen et al., 2019 paper on decreasing mtrCDE efflux pump expression resulting in increasing gonococcal susceptibilty to beta lactam in the discussion section on the contribution of efflux pump to beta lactam MICs. This will help situate the finding compared to other papers in general.

-The phylogenetic tree was used to point out the penA alleles and other resistance determinants; it would be clearer instead to just show the typing, genomic resistance and phenotypic resistance, as no further reference is made to the phylogenetic relationship between the strains in the paper. However, if a phylogenetic tree is desired, the tree constructed needs to take into account homologous recombination between N. gonorrhoeae - it is unclear whether this was considered in the methods section. Alternatively, GrapeTree in PubMLST could be used to examine genomic relationships between the eight isolates.

PLOS authors have the option to publish the peer review history of their article (what does this mean?). If published, this will include your full peer review and any attached files.

Reviewer #1: No

Reviewer #2: No
---

## [Decision Letter · Decision Letter 1]

26 Nov 2024

Dear Dr Moya,

We are pleased to inform you that your manuscript 'Differential contribution of PBP occupancy and efflux on the effectiveness of β-lactams at their target site in clinical isolates of Neisseria gonorrhoeae' has been provisionally accepted for publication in PLOS Pathogens.

Best regards,

Jose Luis Balcazar, Ph.D.

Academic Editor

PLOS Pathogens

D. Scott Samuels

Section Editor

PLOS Pathogens

Michael Malim

Editor-in-Chief

PLOS Pathogens

orcid.org/0000-0002-7699-2064

The authors have addressed all reviewers' comments and made substantial improvements to the manuscript. Many thanks!

Reviewer Comments (if any, and for reference):

Reviewer's Responses to Questions

**Part I - Summary**

Reviewer #1: The work presented in this revised manuscript advances knowledge regarding antibiotic resistance in N. gonorrhoeae. The work was performed with rigor and the overall scholarship is high.

Reviewer #2: (No Response)

**Part II – Major Issues: Key Experiments Required for Acceptance**

Reviewer #1: The authors addressed all of my previous concerns and the additional experimental results and minor edits strengthened the paper.

Reviewer #2: All issues have been addressed

**Part III – Minor Issues: Editorial and Data Presentation Modifications**

Reviewer #1: None

Reviewer #2: All issues have been addressed

PLOS authors have the option to publish the peer review history of their article (what does this mean?). If published, this will include your full peer review and any attached files.

Reviewer #1: **Yes: **William M. Shafer

Reviewer #2: No

---

## [Editor Report · Acceptance letter]

10 Dec 2024

Dear Dr Moya,

We are delighted to inform you that your manuscript, "Differential contribution of PBP occupancy and efflux on the effectiveness of β-lactams at their target site in clinical isolates of Neisseria gonorrhoeae.," has been formally accepted for publication in PLOS Pathogens.

Best regards,

Sumita Bhaduri-McIntosh

Editor-in-Chief

PLOS Pathogens

orcid.org/0000-0003-2946-9497

Michael Malim

Editor-in-Chief

PLOS Pathogens

orcid.org/0000-0002-7699-2064